# Variable Intrinsic Expression of Immunoregulatory Biomarkers in Breast Cancer Cell Lines, Mammospheres, and Co-Cultures

**DOI:** 10.3390/ijms24054478

**Published:** 2023-02-24

**Authors:** Yoel Genaro Montoyo-Pujol, Marta García-Escolano, José J. Ponce, Silvia Delgado-García, Tina Aurora Martín, Hortensia Ballester, Elena Castellón-Molla, Pascual Martínez-Peinado, Sandra Pascual-García, José Miguel Sempere-Ortells, Gloria Peiró

**Affiliations:** 1Research Unit, Dr. Balmis University General Hospital, Alicante Institute for Health and Biomedical Research (ISABIAL), Pintor Baeza 12, 03010 Alicante, Spain; 2Medical Oncology Department, Dr. Balmis University General Hospital, Alicante Institute for Health and Biomedical Research (ISABIAL), Pintor Baeza 12, 03010 Alicante, Spain; 3Gynecology and Obstetrics Department, Dr. Balmis University General Hospital, Alicante Institute for Health and Biomedical Research (ISABIAL), Pintor Baeza 12, 03010 Alicante, Spain; 4Pathology Department, Dr. Balmis University General Hospital, Alicante Institute for Health and Biomedical Research (ISABIAL), Pintor Baeza 12, 03010 Alicante, Spain; 5Biotechnology Department, Immunology Division, University of Alicante, Ctra San Vicente s/n., 03080 San Vicente del Raspeig, Spain; 6Alicante Institute for Health and Biomedical Research (ISABIAL), Pintor Baeza 12, 03010 Alicante, Spain

**Keywords:** breast carcinoma, immune checkpoints, mRNA expression, cell lines, mammospheres, co-culture

## Abstract

Advances in immunotherapy have increased interest in knowing the role of the immune system in breast cancer (BC) pathogenesis. Therefore, immune checkpoints (IC) and other pathways related to immune regulation, such as JAK2 and FoXO1, have emerged as potential targets for BC treatment. However, their intrinsic gene expression in vitro has not been extensively studied in this neoplasia. Thus, we evaluated the mRNA expression of tumor-cell-intrinsic *CTLA-4, PDCD1* (PD1), *CD274* (PD-L1), *PDCD1LG2* (PD-L2), *CD276* (B7-H3), *JAK2*, and *FoXO1* in different BC cell lines, derived mammospheres, and co-cultures with peripheral blood mononuclear cells (PBMCs) by real-time quantitative polymerase chain reaction (qRT-PCR). Our results showed that intrinsic *CTLA-4, CD274* (PD-L1), and *PDCD1LG2* (PD-L2) were highly expressed in triple-negative cell lines, while *CD276* was predominantly overexpressed in luminal cell lines. In contrast, *JAK2* and *FoXO1* were under-expressed. Moreover, high levels of *CTLA-4, PDCD1* (PD1), *CD274* (PD-L1), *PDCD1LG2* (PD-L2), and *JAK2* were found after mammosphere formation. Finally, the interaction between BC cell lines and peripheral blood mononuclear cells (PBMCs) stimulates the intrinsic expression of *CTLA-4, PCDC1* (PD1), *CD274* (PD-L1), and *PDCD1LG2* (PD-L2). In conclusion, the intrinsic expression of immunoregulatory genes seems very dynamic, depending on BC phenotype, culture conditions, and tumor-immune cell interactions.

## 1. Introduction

Breast cancer (BC) is the most common cancer in women and the leading cause of cancer-related death in this gender worldwide [1]. Although clinical management has improved over the last few years, the efficacy of traditional therapeutic strategies such as surgery, radiation, endocrine therapy, and chemotherapy depends on the patient and the intrinsic tumor subtype. The latter supports that BC is a heterogeneous disease and explains why it behaves more aggressively, leading to recurrences, drug resistance, metastasis, and a worse prognosis [2,3,4]. Therefore, there is a need to identify new therapeutic targets to improve BC patients’ treatment and outcomes.

Historically, BC has been considered a poorly immunogenic neoplasm compared to others with a higher mutational load [5]. However, in recent years, there has been growing interest in knowing the immune system’s role in the pathogenesis of this neoplasia. In this regard, considering the advances made with immunotherapy, immune checkpoints (ICs) and pathways related to regulating the immune response have emerged as potential therapeutic targets for BC. The IMpassion130 clinical trial demonstrated the benefit of atezolizumab plus nab-paclitaxel in the progression-free survival (PFS) and the overall survival (OS) of patients with locally advanced or metastatic triple-negative BC with programmed cell death ligand-1 (PD-L1) immune cell-positive (>1%) disease [6,7,8,9]. In the KEYNOTE-355 clinical trial, an improvement in progression-free survival (PFS) was also observed with pembrolizumab plus chemotherapy in untreated patients with locally recurrent inoperable or metastatic triple-negative BC whose tumors expressed PD-L1 with a combined positive score (CPS) ≥ 10 [10]. Furthermore, the KEYNOTE-522 clinical trial, performed on previously untreated stage II or III triple-negative BC has reported an increase in event-free survival (EFS) among patients who received pembrolizumab plus neoadjuvant chemotherapy [11]. Based on these findings, atezolizumab plus nab-paclitaxel and pembrolizumab in combination with chemotherapy were approved by the Food and Drug Administration (FDA). However, the discrepant results of the IMpassion131 trial, where the combination of atezolizumab with paclitaxel did not improve PFS or overall survival (OS) versus paclitaxel alone [12], prompted the FDA to warn healthcare professionals not to substitute nab-paclitaxel when combined with atezolizumab for the treatment of metastatic triple-negative BC patients. Clinical trials with anti-cytotoxic T lymphocyte antigen 4 (CTLA-4) agents are still in progress, and very few data are available. Nevertheless, promising results have been obtained in a phase I trial using tremelimumab (anti-CTLA-4) in combination with local radiation. The stable disease has been the best response in patients with inoperable locally recurrent or metastatic BC [13]. A single-arm pilot study investigating the combination of durvalumab (anti-PD-L1) with tremelimumab has also reported an objective response rate (ORR) of 43% in patients with metastatic triple-negative BC, with no responses in luminal tumors [14]. However, the authors of these studies state that their results require validation. Therefore, more research is needed to understand better the role of the immune system in BC, particularly the role of ICs.

ICs are a set of proteins that modulate the duration and amplitude of the immune response. They are a set of proteins that maintain homeostasis and self-tolerance under physiological conditions. However, within a pathological context, tumor cells can express these molecules as an essential mechanism of immune resistance [15]. The ICs that have been studied the most are CTLA-4 and programmed cell death protein 1 (PD1), as well as their corresponding ligands, PD-L1 and PD-L2 [16]. In addition, due to advances made in recent years, B7-H3 (CD276) has been added as a new protein to the IC family [17].

CTLA-4 is mainly expressed on activated, memory, and regulatory T cells (Tregs) [18] and is responsible for inhibiting effector T cells. However, it is not only expressed in immune cells but also neoplastic cells [19]. Furthermore, its soluble form has also been detected in the serum of patients with cancer [20] and autoimmune disorders [21], suggesting that CTLA-4 may be involved in functions beyond the immune system regulation.

On the other hand, PD1 is mainly expressed on the surface of activated T cells and B cells, natural killer (NK) cells, activated monocytes, dendritic cells, and thymocytes [16]. Its principal function is to negatively regulate co-stimulatory signals, inhibiting cell proliferation, cytokine production, and cytotoxic activity [16] through interaction with PD-L1 and PD-L2 and the recruitment of SHP1 and SHP2 phosphatases [22]. Recent studies have demonstrated PD1 intrinsic expression in different neoplasms and tumor cell lines [23,24]. PD-L1 is widely expressed in various cells, such as antigen-presenting cells (APCs), T and B lymphocytes, monocytes, and epithelial cells. On the other hand, PD-L2 is mainly restricted to APCs, especially dendritic cells and macrophages, but it can also be induced in mast cells and B lymphocytes [16]. Like other ICs, they can also be intrinsically expressed in neoplastic cells. In addition, they also participate through intrinsic functions in the proliferation, migration, and invasion of neoplastic cells [25,26], regulation of epithelial-to-mesenchymal transition (EMT) [26,27], acquisition of stem cell characteristics [27,28,29], inhibition of apoptosis, and resistance to chemotherapy [30]. 

B7-H3 (CD276) is mainly expressed in APCs [31,32], activated T and B cells [31], NK cells [33], fibroblasts, epithelial and endothelial cells [34], and in a variety of malignancies [35,36,37]. Although initially identified as a co-stimulatory molecule [38], it was later described as a co-inhibitory molecule in different neoplasias [39]. Moreover, it has been said to enhance tumor aggressiveness through non-immunological functions [40].

Of note, ICs are not the only proteins that regulate the immune response. Other molecules, such as JAK2 and FoXO1, also play an essential role. Thus, JAK2 is a tyrosine kinase protein that, together with signal transducer and activator of transcription (STATs) proteins, is responsible for initiating the transcription of different target genes involved in various cellular processes [41]. In addition, JAK2 has also been implicated in activating critical signaling pathways such as the PI3K/Akt or RAS/RAF/MAPK [42]. Its expression in neoplastic cells has been associated with poor prognosis and clinicopathological features [43,44].

Further, it has been directly involved in cell proliferation, migration and invasion, angiogenesis, resistance to treatment and apoptosis, and the maintenance of stemness and EMT in different neoplasms [45,46,47]. It is also essential for PD-L1 and PD-L2 expression in neoplastic cells and for promoting non-immunological functions after interacting with B7-H3 [37,48,49]. On the other hand, FoXO1 is a protein belonging to the FoXO subfamily of transcription factors, which also includes the FoXO3, FoXO4, and FoXO6 proteins [50]. Their primary function is to transcribe different genes involved in various physiological processes [51], such as the migration of dendritic cells [52], the production of inflammatory mediators, and the polarization of monocyte–macrophage cells [53], the maturation of B cells, and Tregs [54], as well as the formation and subsequent function of memory T cells [55]. Likewise, its expression in neoplastic cells is an independent factor of good prognosis [56] and limits the migration and invasion of neoplastic cells by inhibiting the EMT process [57]. Thus, FoXO1 is considered by many authors as a tumor suppressor protein. However, there are contradictory data since, depending on the type of neoplasm and the tumor microenvironment, it could also act as an oncogene, promoting cell proliferation and drug resistance in different neoplasms [58,59].

Therefore, there is a need to analyze the role of ICs, JAK2, and FoXO1 in regulating the immune response. Currently, the potential involvement of these molecules in the pathogenesis of BC has not been extensively investigated in different phenotypes, and data from in vitro studies and stem cell conditions are limited. 

We hypothesize that intrinsic expression of *CTLA-4, PDCD1* (PD1), *CD274* (PD-L1), *PDCD1LG2* (PD-L2), *CD276* (B7-H3), *JAK2*, and *FoXO1* in neoplastic cells depends on BC immunophenotype, stem cell properties, and tumor microenvironment. Therefore, we conducted different in vitro studies on six BC cell lines, derived mammospheres, and co-cultures with peripheral blood mononuclear cells (PBMCs) to determine this question. 

Our data showed that *CTLA-4, CD274* (PD-L1), and *PDCD1LG2* (PD-L2) were intrinsically overexpressed in triple-negative cell lines, while *CD276* was increased in luminal cell lines, both in the absence of adaptative immunity. In contrast, *JAK2* and *FoXO1* were under-expressed in most of the cell lines analyzed. Moreover, the expression of *CTLA-4, PDCD1* (PD1), *CD274* (PD-L1), *PDCD1LG2* (PD-L2), and *JAK2* in aggressive phenotypes increased after mammosphere formation. Finally, we observed that the interaction between BC cell lines and PBMCs stimulates the intrinsic expression of *CTLA-4, PCDC1* (PD1), *CD274* (PD-L1), and *PDCD1LG2* (PD-L2), which may vary depending on BC immunophenotype and time of interaction.

## 2. Results

### 2.1. Intrinsic Gene Expression in BC Cell Lines

We detected that healthy mammary epithelial cell line 184A1 did not express *CTLA-4* and *PDCD1* (PD1) receptors, so the ΔCt value was used for the analysis. In this case, the ΔCt value acted as a relative gene expression value, where low values are indicative of higher gene expression. We observed a significant increase in *CTLA-4* in MDA-MB-231 and MDA-MB-468 compared to MCF-7 (both *p* = 0.025) and BT-474 (*p* = 0.009 and *p* = 0.024, respectively), but not against SK-BR-3 (*p* = 0.193 and *p* = 0.385, respectively). In addition, *CTLA-4* expression was not detected in T-47D (Figure 1A). Regarding *PDCD1* (PD1), the expression remained stable in all cell lines, with no significant differences found among them (all *p* > 0.05) (Figure 1A).

*CD274* (PD-L1) and *PDCD1LG2* (PD-L2) were also increased in MDA-MB-231 compared to 184A1, which was only statistically significant for *CD274* (PD-L1) (*p* = 0.019). In the other cell lines studied, both genes were found to be significantly under-expressed (all *p* < 0.014) (Figure 1B). In addition, it is important to highlight the absence of *PDCD1LG2* (PD-L2) expression in BT-474. Among the tumor cell lines, MDA-MB-231 presented the highest levels of *CD274* (PD-L1) and *PDCD1LG2* (PD-L2) expression compared to all cell lines (all *p* < 0.014). Likewise, an under-expression of both genes was observed in MCF-7, T-47D, and BT-474 cell lines concerning SK-BR-3 (all *p* < 0.020).

*CD276* (B7-H3) expression was significantly higher in MCF-7 (*p* = 0.049) and BT-474 (*p* = 0.021) cell lines. A significant decrease was also observed in SK-BR-3 (*p* = 0.003) compared to normal cell line 184A1 (Figure 1C). When comparing the neoplastic lines, a significant increase was detected in MCF-7 and BT-474 compared to SK-BR-3 (all *p* < 0.028).

On the other hand, *JAK2* was significantly under-expressed in MCF-7, T-47D, BT-474, and SK-BR-3 cell lines (all *p* < 0.024) compared with 184A1. A slight decrease was also seen for MDA-MB-231, although only as a trend towards significance (*p* = 0.055) (Figure 1C). Furthermore, when compared with each other, it was observed that *JAK2* expression was significantly decreased in BT-474 compared to MCF-7 (*p* = 0.022), SK-BR-3 (*p* = 0.012), and MDA-MB-468 (*p* = 0.021) cell lines. Finally, *FoXO1* was significantly under-expressed in BT-474, SK-BR-3, and MDA-MB-231 (all *p* < 0.015) cell lines compared with 184A1 (Figure 1C). Among the tumor cell lines, BT-474 presented the lowest expression (all *p* < 0.05).

### 2.2. Intrinsic Gene Expression in Mammospheres

Different BC cell lines were cultured under non-adherent conditions to obtain mammospheres. Enriched in cancer stem cells, these spheres are characterized by a *CD44^high^/CD24^low^* phenotype in BC [60]. Therefore, both gene expression levels were measured to verify that the generation of mammospheres had been performed correctly. In this regard, all cases showed increased *CD44* expression and normal or decreased *CD24* expression compared to controls (Appendix A). However, we failed to generate mammospheres from the MDA-MB-468 cell line, so it was not considered for further analysis. Once *CD44* and *CD24* expression were verified, the relative gene expression of our genes of interest was quantified. 

Regarding *CTLA-4* and *CD274* (PD-L1) expression, we observed that both genes were significantly overexpressed in the mammospheres of BT-474 and SK-BR-3 cell lines (all *p* < 0.029) compared to controls. This increase also occurred in the mammospheres of the MCF-7 cell line but was not statistically significant. In addition, an increase in *CTLA-4* expression (*p* < 0.029) and a decrease in *CD274* (PD-L1) expression (*p* < 0.05) were observed in MDA-MB-231 mammospheres. Finally, in T-47D mammospheres, *CD274* (PD-L1) expression was significantly under-expressed (*p* < 0.05), while *CTLA-4* was not detected (Figure 2A). After comparing the mammospheres with each other, we found that those derived from the MDA-MB-231 line showed the highest levels of *CTLA-4*. However, this difference was only significant concerning SK-BR-3 mammospheres (*p* = 0.035). On the other hand, *CD274* (PD-L1) was significantly under-expressed in T-47D and MDA-MB-231 mammospheres concerning the rest (in all cases, *p* < 0.042).

Regarding *PDCD1* (PD1) and *PDCD1LG2* (PD-L2) expression, there were statistically significant differences between mammospheres obtained from MCF-7 and MDA-MB-231 cell lines (all cases *p* < 0.037) and controls. However, the levels remained close to normal in the rest of the mammospheres (Figure 2B). Furthermore, as in the control cells, *PDCD1LG2* (PD-L2) was not expressed in BT-474 mammospheres. When comparing the mammospheres, it was seen that *PDCD1* (PD1) expression in MDA-MB-231 mammospheres was significantly higher than in the rest (all *p* = 0.021). At the same time, *PDCD1LG2* (PD-L2) levels were significantly lower in SK-BR-3 mammospheres versus those derived from MCF-7 and MDA-MB-231 (both *p* < 0.018).

As can be seen in Figure 2C, *CD276* (B7-H3) was under-expressed in mammospheres obtained from BT-474, T-47D, and SK-BR-3 cell lines, being significant in the latter two cases (both *p* < 0.025) compared to controls. On the other hand, the remaining mammospheres showed *CD276* (B7-H3) levels close to the controls. Likewise, it was observed that the mammospheres derived from the MDA-MB-231 cell line presented a significantly higher expression (all *p* = 0.05) than the rest, except those derived from MCF-7 (*p* = 0.513).

*JAK2* expression was remarkably under-expressed in T-47D and BT-474 mammospheres (both *p* = 0.014) and over-expressed in SK-BR-3 and MDA-MB-231 (both *p* = 0.014) compared to the controls (Figure 2C). Furthermore, when comparing the results, we found that mammospheres derived from SK-BR-3 and MDA-MB-231 cell lines had significantly higher *JAK2* levels than the rest (all cases *p* = 0.021). 

Finally, *FoXO1* remained statistically underexpressed in most mammospheres (all cases *p* < 0.026) compared to controls. The only exception was the MDA-MB-231 mammospheres that maintained a near-normal expression (Figure 2C). After reaching the results between mammospheres, we found that those derived from the MDA-MB-231 cell line showed a significant overexpression (all *p* < 0.033). In contrast, those derived from T-47D showed a significant underexpression (all *p* < 0.020).

### 2.3. Intrinsic Gene Expression in Co-Cultures of BC Cell Lines and PBMCs

We developed a co-culture model with PBMCs derived from BC patients and commercial cell lines, representing the different immunophenotypes. We aimed to determine the effect of direct interaction between tumor cells and immune cells on the intrinsic expression of immunoregulatory gene expression. *CTLA-4* and *PDCD1* (PD1) increased expression was seen in all co-cultures compared with control cell lines (Figure 3A,B). No significant differences in *CTLA-4* expression were found for any co-culture on the different days the analysis was carried out. However, we observed that the T-47D cell line expressed this gene after interaction with PBMCs, thus representing the ΔCt values (Figure 3A). Regarding *PDCD1* (PD1), the differences in expression observed between the second and third day in the T-47D cell line and those between the first and second day for the MDA-MB-468 cell line were statistically significant (both *p* = 0.043) (Figure 3B).

Similar results were observed for *CD274* (PD-L1) and *PDCD1LG2* (PD-L2), showing an increase (Figure 3C,D). Likewise, we found significant differences in *CD274* (PD-L1) levels during the first day of interaction with PBMCs in the SK-BR-3 cell line compared to the second and the third day on which the measurements were carried out (*p* = 0.037 and 0.027, respectively) (Figure 3C). On the other hand, *PDCD1LG2* (PD-L2) was significantly elevated on the second day of interaction for the T-47D and MDA-MB-231 cell lines compared with the first day of measurements (both *p* = 0.043) (Figure 3D). In addition, we observed the expression of *PDCD1LG2* (PD-L2) in BT-474 cells, with significant differences between the ΔCt values on the second and third day (*p* = 0.043). In contrast, *CD276* (B7-H3) and *JAK2* levels in co-cultures did not differ from those obtained in the cell lines used as controls (Figure 3E,F). However, in the case of the MDA-MB-468 cell line, a significant increase of both genes was observed 72 h after interaction with PBMCs compared to the other days (all *p* < 0.027). Finally, *FoXO1* expression remained at near-normal levels in all co-cultures. Nevertheless, we observed significant differences between the second and the third day of interaction in the co-cultures of the MCF-7 cell line (*p* = 0.030). We also found differences between the first and the second day (*p* = 0.002), as well as the second and the third day of interaction (*p* = 0.036) in the co-cultures of BT-474 (Figure 3G).

## 3. Discussion

In the current study, we analyzed the intrinsic mRNA expression of several immunoregulatory genes, such as *CTLA-4, PDCD1* (PD1), *CD274* (PD-L1), *PDCD1LG2* (PD-L2), *CD276* (B7-H3), *JAK2* and *FoXO1* in six BC cell lines, their derived mammospheres, and in co-cultures with PBMCs. Our data showed an intrinsic *CTLA-4* mRNA increase in the triple-negative cell lines MDA-MB-231 and MDA-MB-468 compared to the luminal cell lines MCF-7 and BT-474. This increase was also observed in mammospheres derived from BT-474, SK-BR-3, and MDA-MB-231 cell lines. Moreover, as in the controls, no expression was detected in the luminal A T-47D cell line and derived mammospheres. In agreement with our results, previous studies have also demonstrated its expression in MCF-7 and MDA-MB-231 BC cell lines at both mRNA and protein levels in the absence of immune system cells [19,61,62,63], but not in BT-474, SK-BR-3, and MDA-MB-468 cell lines. Likewise, it has been shown that its expression in neoplastic BC cells can suppress dendritic cell maturation in vitro [61]. In addition, an independent study confirmed that CTLA-4-CD80 binding is strong enough to internalize CD80 inside the neoplastic cells, thus attenuating the ability of APCs to activate T lymphocytes [63].

Intrinsic CTLA-4 expression has also been linked to BC cell proliferation, differentiation, survival [62], and the stem-cell-like phenotype in different neoplasms [64]. Of note, its essential role in maintaining pluripotency and the ability to form mammospheres in vitro has been previously described in melanoma cells [65]. Nevertheless, cancer-cell-intrinsic *CTLA-4* expression has never been previously analyzed in mammospheres. According to the current results, *CTLA-4* expression on neoplastic cells may be essential for the pathogenesis of BC and the modulation of the tumor microenvironment, especially in triple-negative BC. In addition, our results suggest that its expression might be implicated in acquiring and maintaining tumor stem cell characteristics in aggressive BC phenotypes. 

We also demonstrated that *PDCD1* (PD1) maintained a similar mRNA level for all analyzed cell lines, with no significant differences. However, it was overexpressed in mammospheres derived from MCF-7 and MDA-MB-231 cell lines. This increase was significant in the rest of the MDA-MB-231 mammospheres. To our knowledge, this is the first study to characterize the intrinsic expression of *PDCD1* (PD1) in BC cell lines and mammospheres without an immunological environment. Of note, its expression has been evaluated in other neoplasms, where in most cases, it has been described as a pro-oncogenic factor. In this regard, several independent studies have confirmed that the intrinsic expression of PD1 in melanoma, hepatocellular carcinoma, and bladder cancer cell lines promotes neoplastic growth without an adaptative immune response [66,67,68,69]. Likewise, its overexpression in pancreatic ductal adenocarcinoma cell lines was associated with increased tumor growth after cell injection in immunosuppressed mice [70]. In contrast, PD1 overexpression in the murine cell line M109 showed a significant decrease in viability and cell proliferation [71]. These data demonstrate that the intrinsic expression and function of PD1 in neoplastic cells are complex and not yet fully defined. 

*CD274* (PD-L1) and *PDCD1LG2* (PD-L2) were intrinsically overexpressed in the triple-negative MDA-MB-231 cell line, although only significant for CD274 (PD-L1). The rest of the cell lines showed underexpressed levels compared to controls. In addition, the absence of *PDCD1LG2* (PD-L2) expression in the BT-474 line was noteworthy. In agreement with our results, numerous authors have observed a significant increase of both ligands in the MDA-MB-231 cell line compared with luminal cell lines, both at the mRNA and protein levels [72,73,74,75]. On the other hand, in partial agreement, decreased mRNA *PDCD1LG2* (PD-L2) has also been reported in MDA-MB-231, SK-BR-3, and MCF-7 compared to the non-malignant MCF10A cell line by others [76]. Therefore, it can be stated that the expression of both genes varies considerably between the different BC phenotypes. Similarly, there is evidence that these genes are expressed synchronously in neoplastic cells [75], and there may be changes in response to endogenous and exogenous factors [72]. 

Concerning intrinsic functions, PD-L2 has been less studied in cell lines, whereas PD-L1 has been implicated in resistance to chemo- and radiotherapy in the MDA-MB-231 cell line [30] and apoptosis, as well as induction of cell proliferation, migration, and invasion processes in gastric cancer and pancreatic carcinoma cell lines [25]. These data support the idea that intrinsic *CD274* (PD-L1) expression in neoplastic cells favors immune response evasion and increases neoplastic aggressiveness. Their gene expression levels have also been related to tumor stemness maintenance in different neoplasms [27,77]. We found that intrinsic *CD274* (PD-L1) was significantly overexpressed in mammospheres derived from HER2-expressing cell lines (BT-474 and SK-BR-3). In contrast, compared with controls, intrinsic *PDCD1LG2* (PD-L2) was upregulated in those derived from MCF-7 and MDA-MB-231. In agreement with our results, previous investigators demonstrated intrinsic PD-L2 expression by flow cytometry in MCF-7-derived mammospheres with no significant differences concerning the control cell line [78].

In contrast, a high intrinsic PD-L1 expression at the mRNA and protein level has been described in mammospheres derived from MCF-7 and MDA-MB-231 cell lines [29,78]. Furthermore, its expression has been reported to be necessary to maintain the expression of transcription factors related to stem cell phenotypes such as OCT-4A, Nanog, and BMI1 [28,29]. Likewise, an in silico study performed in the MDA-MB-231 cell line revealed an association between intrinsic PD-L1, the EGFR signaling pathway, and several factors involved in the EMT process and tumor stemness [79]. At the same time, other investigators have related the expression with the activation of Notch and/or PI3K/AKT signaling pathways [80]. Therefore, our data further support the essential role of intrinsic *CD274* (PD-L1) and *PDCD1LG2* (PD-L2) in maintaining BC’s tumor stem cell phenotype.

We detected that intrinsic *CD276* (B7-H3) was overexpressed in luminal cell lines (MCF-7 and BT-474) and underexpressed in HER2-enriched (SK-BR-3) compared to the control. In the remaining cell lines, the expression was close to normal. Conversely, BT-474, T-47D, and SK-BR-3-derived mammospheres showed decreased expression, while those derived from MCF-7 and MDA-MB-231 presented levels close to normal. The intrinsic expression of this immune checkpoint has been previously described in MCF-7 and T-47D cell lines in variable levels [31]. Furthermore, regarding its function, it has been observed that silencing its expression in the MCF-7 cell line results in an increase in vascular endothelial growth factor (VEGF), both at the mRNA and protein levels [33]. These results suggest that intrinsic B7-H3 may be involved in controlling neoplastic cell growth. However, independent studies confirm that its expression in the MDA-MB-231 cell line not only favors tumor growth but also promotes the Warburg effect [81] and induces resistance to paclitaxel [82] and inhibitory molecules of the PI3K/AKT/mTOR pathway [83] through activation of JAK2/STAT3. Therefore, all these data reflect that the function of intrinsic B7-H3 is not yet fully defined in BC, as it appears to play a dual role in the pathogenesis of the molecular subtypes. 

Furthermore, recent studies have shown that intrinsic *CD276* (B7-H3) expression in neoplastic cells plays a crucial role in the induction of the EMT process and acquiring stem cell phenotype. For example, its intrinsic expression in colorectal cancer cell lines decreased E-cadherin expression and increased stem cell-associated markers such as N-cadherin, vimentin, CD133, CD44, and OCT4 [84]. Likewise, other authors have demonstrated its association with neural stem cell markers in glioblastomas [36] and the capacity to form tumorspheres [85]. In addition, there is evidence that intrinsic *CD276* (B7-H3) is a critical factor in evading the immune response by tumor stem cells in head and neck carcinoma [86]. In BC, this gene has been associated with gene signatures related to increased cell invasion and expression of stem cell markers and signaling pathways involved in pluripotency [87]. Indeed, an independent study demonstrated that B7-H3 expression increased the number of BC stem cells through the major vault protein (MVP), thus promoting tumor development [35]. In contrast, our data do not support that B7-H3 is directly involved in maintaining the tumor stem cell properties in BC cell lines.

Intrinsic *JAK2* was significantly under-expressed in the luminal (MCF-7, T-47D, and BT-474), HER2-enriched (SK-BR-3), and triple-negative MDA-MB-231 cell lines, the latter only as a trend. However, in MDA-MB-468, cell line values were close to normal. Balko et al. suggest that the difference between the MDA-MB-231 and MDA-MB-468 cell lines could be explained by variations in the gene copy number [88]. They also found overexpression of *JAK2* in the MCF-7 cell line, in contrast to our data.

Regarding its function, it is directly involved in the proliferation and aggressiveness of MDA-MB-231 and MDA-MB-468 cell lines [89]. Likewise, it would be involved in angiogenesis, cell proliferation, migration, and invasion in the tamoxifen-resistant (TAMR)MCF-7 cell line [47]. Concerning mammospheres, significantly decreased levels were observed in those derived from T-47D and BT-474, with opposite results for SK-BR-3 and MDA-MB-231. In agreement, prior studies have detected that JAK2 inhibition in the MDA-MB-231 cell line reduces the number of CD44^high^/CD24^low^ cells and inhibits the formation of mammospheres [90,91]. Similar results were also seen in the cell lines MCF-7/ADR, resistant to doxorubicin, and MCF-7/SC, a cell line that, compared to MCF-7, shows high invasiveness and stem cell properties [91,92]. However, it should be noted that these results are not new, as previously, Marotta et al. demonstrated the involvement of the IL-6/JAK2/STAT3 axis in the growth and survival of BC stem cells [93]. Moreover, JAK2 activation is involved in Twist-1 and Twist-2 overexpression, related to the EMT process and the acquisition of stem cell phenotype [94]. In like manner, increased copy number or amplification of *JAK2* in triple-negative cell lines favored mammospheres’ formation and increased chemotherapy resistance [88]. This property was also observed after activating the JAK2/STAT3 pathway via PAK1 [95]. Thus, these data suggest that *JAK2* expression is essential for mammosphere formation, maintenance of the stem cell phenotype in BC, and resistance to conventional treatments.

We demonstrated an intrinsic underexpression of *FoXO1* in the most aggressive cell lines (BT-474, SK-BR-3, and MDA-MB-231). In addition, Liang et al. reported in the MDA-MB-231 cell line that it promotes cell proliferation but not migration or apoptosis [96]. However, luminal A cell lines (MCF-7 and T-47D) and MDA-MB-468 showed expression values close to normal. Of note, Gutilla et al. have found in the MCF-7 a large discrepancy between *FoXO1* mRNA and protein levels, suggesting a robust translational control [97].

We also detected a decrease in *FoXO1* expression in all analyzed mammospheres, except those from the MDA-MB-231 cell line. Similar results have been previously reported in MCF-7, SK-BR-3, and MDA-MB-231 [96,98,99]. Despite this, FoXO1 inhibits EMT and metastasis in the MCF-7 cell line [99]. Moreover, it is involved in the resistance mechanisms of tamoxifen [100] and doxorubicin [101] in TAMR-MCF-7 and MCF-7/ADR cell lines, respectively. In line with recent studies, we showed a positive correlation between FoXO1 underexpression and mammosphere formation in the MCF- [102].

In contrast, Truong et al. demonstrated that *FoXO1* expression in MCF-7, T-47D, and BT-474 was directly related to progesterone receptor phosphorylation, inhibition of cell proliferation, and mammosphere formation [103]. These contradictory results may be because the cell lines were supplemented with hormones. Nevertheless, discrepancies remain regarding the function of this transcription factor. Indeed, another independent research showed that FoXO1 accumulation in BC cells promoted the expression of SOX2, a transcription factor related to the tumor stem cell phenotype [104]. The previous data suggest that *FoXO1* may play a critical role in BC pathogenesis and drug resistance development. Although the published data are not entirely clear, our results show the relevance of *FoXO1* in the maintenance of the tumor stem cell phenotype. 

Finally, we observed in all co-cultures a significant increase in *CTLA-4, PCDC1* (PD1), *CD274* (PD-L1), and *PDCD1LG2* (PD-L2) intrinsic expression after a direct interaction with PBMCs. Furthermore, T-47D and BT-474 cell lines recovered cell-intrinsic *CTLA-4* and *PDCD1LG2* (PD-L2) expression after tumor-immune cell interactions, respectively. However, *CD276* (B7-H3), *JAK2,* and *FoXO1* maintained their intrinsic expression levels close to normal, even after interaction with PBMCs. Similar results were already reported for PD-L1 by Hinterneder et al. [105], who described that direct interaction of the HCC38 cell line with activated PBMCs increased intrinsic PD-L1 protein expression. Likewise, co-culture of melanoma metastasis and their autologous TILs induced the expression of several immunomodulatory molecules, including PD-L1 or indoleamine 2,3-dioxygenase (IDO), after 24 h of interaction [106]. In this regard, these gene expressions in neoplastic cells can respond to various cytokines and growth factors in the tumor microenvironment. Indeed, IFN-γ, produced by TILs after antigenic recognition, is one of the most potent extrinsic inducers for PD-L1 and PD-L2 expression in neoplastic cells [107]. Furthermore, intrinsic PD1 expression in ovarian cancer in response to IFN-α and IFN-γ has also been described [69], although these results could not be confirmed in non-small-cell lung cancer [23]. For its part, intrinsic CTLA-4 expression in melanoma cells would also be IFN-γ-dependent [108]. Despite this, a recent study has shown that the production of soluble factors is insufficient to inhibit immune cells, which are necessary for the interaction between tumor and stromal cells [109]. Here, we demonstrated the relevance of the interaction between neoplastic and immune cells in inducing the intrinsic expression of several immunoregulatory genes as an adaptive mechanism against active antineoplastic response. Our results also showed a time-dependent variation in the BC cell line’s intrinsic gene expression. In fact, for the MCF-7 cell line, *FoXO1* expression increased at 72 h of interaction compared to the first 48 h. Additionally, the T-47D cell line showed an increase in *PDCD1* (PD1) and *PDCD1LG2* (PD-L2) on the second day compared to the first and the third day, respectively. Likewise, the BT-474 cell line experienced an increase in *PDCD1LG2* (PD-L2) levels at 72 h of interaction, while *FoXO1* was overexpressed after the first 24 h. Regarding the MDA-MB-468 cell line, *PDCD1* (PD1) was under-expressed at 48 h versus 24 h, while *CD276* (B7-H3) and *JAK2* were over-expressed at 72 h of interaction. Finally, the MDA-MB-231 cell line only manifested an increase in *PDCD1LG2* (PD-L2) during the second interaction day compared to the first one. These temporal variations in intrinsic gene expression could be due to the inhibition of immune cells and thus less production of soluble factors and fewer cell-cell interactions. In support of this, Chen et al. demonstrated in BC cell lines that the intrinsic expression of CTLA-4 could inhibit the processes of antigenic presentation and expression of inflammatory cytokines, preventing the proper activation of type 1 T helper (Th1) and cytotoxic T lymphocytes [61].

Similarly, overexpression of PD-L1 and PD-L2 in neoplastic cells would directly inhibit effector T lymphocytes [16]. Our experimental data suggest that intrinsic expression of the immunoregulatory genes is dynamic and highly dependent on the BC cell line subtypes and the inflammatory context. Nevertheless, these results are not possible to corroborate in clinical series. Therefore, co-cultures are a valuable tool for better understanding all these processes. However, further in vivo studies are necessary to confirm our results and elucidate immunoregulatory genes’ role in BC pathogenesis.

## 4. Materials and Methods

### 4.1. BC Cell Lines Culture 

Six human BC cell lines, MCF-7 and T-47D (luminal A-like), BT-474 (luminal B/HER2-positive), SK-BR-3 (HER2-enriched), MDA-MB-231 and MDA-MB-468 (triple-negative/basal-like), and the normal epithelial breast cell line 184A1 were obtained from the American Type Culture Collection (ATCC) (Manassas, VA, USA) (Figure 4). All cell lines were maintained in a 75 cm^2^ flask (SPL Life Sciences, Gyeonggi, Korea) and Dulbecco Modified Eagle Medium (DMEM)/Ham F12 (1:1) with L-glutamine and 15 mM HEPES media (Biowest) supplemented with 10% fetal bovine serum (FBS) (Biowest), 50 U/mL of penicillin, and 50 mg/mL of streptomycin (Biowest). Cells were incubated at 37 °C in a humidified atmosphere with 5% of CO_2_ and were grown to around 80% confluence before RNA isolation.

### 4.2. Isolation of Human PBMCs

Peripheral blood was collected from 30 BC patients with informed consent to use their samples for research purposes and ethical approval from the ethics committee (ethic code PI2019/089 and PI2020-265) at our institution. The selected patients represented each BC immunophenotype (10 luminal A, five luminal B/HER2-positive, 5 HER2-enriched, and ten triple-negative patients). PBMCs were isolated by density gradient centrifugation. Briefly, 3 mL of Biocoll Separating Solution (Biochrom Ltd., Cambridge, United Kingdom) and 6 mL of peripheral blood samples diluted in phosphate-buffered saline (PBS) were placed in sterile tubes centrifuged at 1300× *g* for 15 min. Further, the PBMCs were collected from the interface between plasma and Biocoll, washed twice, and cultured in Roswell Park Memorial Institute (RPMI) 1640 medium with glutamine (Capricorn Scientific GmbH, Ebsdorfergrund, Germany) and supplemented with 10% FBS (Capricorn Scientific GmbH) and 1% antibiotics (50 U/mL penicillin and 50 mg/mL streptomycin) (Capricorn Scientific GmbH). PBMCs were incubated for 2 h at 37 °C and 5% CO_2_, allowing monocyte separation by adhering to the plastic of the culture flask [110]. After this time, PBMCs were collected and assessed for viability using 0.4% trypan blue. Finally, PBMCs were brought to the desired concentration for co-culturing.

### 4.3. Co-Culture of PBMCs with BC Cell Lines

For co-cultures, all tumor cell lines were cultured in RPMI 1640 medium with glutamine (Capricorn Scientific GmbH), supplemented with 10% FBS (Biowest), 50 U/mL of penicillin, and 50 mg/mL of streptomycin (Biowest). The cell lines were seeded in sterile culture flasks and incubated in a humid atmosphere at 37 °C and 5% CO_2_. Once cultures reached 70–80% confluence, tumor cells were seeded in duplicate at 100,000 cells/well in sterile 6-well plates (SPL Life Sciences). After 24 h, the isolated PBMC were co-cultured in the 6-well plates with the pre-seeded tumor cells at a concentration of 500,000 cells/well. The above was performed considering the phenotype of the cell lines and the BC patients who donated the peripheral blood sample. The co-cultures were then maintained in a humid environment at 37 °C and 5% CO_2_ (Figure 5). After co-culture for 24, 48, and 72 h, the supernatant with PBMCs was collected, and tumor cell lines were washed twice with PBS (Biowest). Before RNA isolation, tumor cells were detached using trypsin-ethylenediaminetetraacetic acid (EDTA) 0.05% (Capricorn Scientific GmbH), washed, and resuspended in PBS.

### 4.4. Mammosphere Culture 

Mammosphere culture of human BC cell lines were performed using a commercial MammoCultTM medium (Stemcell Technologies, Vancouver, BC, Canada) supplemented with 10% proliferation supplements (Stemcell Technologies), 0.48 mg/mL of hydrocortisone (Stemcell Technologies), 4 mg/mL of heparin (Stemcell technologies), 50 U/mL of penicillin, and 50 mg/mL of streptomycin (Biowest). Single T-47D, MCF-7, BT-474, SK-BR-3, and MDA-MB-231 cells were seeded in six-well plates coated with poly-2-hydroxyethyl-methacrylate (Sigma-Aldrich, St Louis, MO, USA) to prevent cell attachment at a density of 5 × 10^3^ cells per well. Under these conditions, the cells grew as a non-adherent spherical cluster (Figure 4). Every two days, 2 mL of fresh medium was added to each well. After seven days in culture, mammospheres were harvested by gentle centrifugation and enzymatically dissociated with trypsin-EDTA 0.05% in PBS w/o calcium w/o magnesium w/o phenol (Biowest) before RNA isolation. All experiments were performed in triplicate.

### 4.5. RNA Isolation and Complementary DNA Synthesis

Single-cell suspensions obtained from cell line cultures, co-cultures, and mammospheres were used for RNA isolation. Cells were lysed in 600 µL of RLT buffer (Qiagen, Hilden, Germany) supplemented with 1% β-mercaptoethanol (Merk Schuchardt OHG, Hohenbrunn, Germany) through a 29G needle. Following the RNeasy Mini kit protocol (Qiagen,), 70% ethanol was added to cell lysates, and the samples were mixed by pipetting before being transferred to the columns. Total RNA was extracted according to the manufacturer’s protocol with on-column DNase digestion. RNA concentration and purity were assessed using a NanoDrop spectrophotometer (Thermo Fisher Scientific, Waltham, MA, USA) and optical density ratios of 260/230 nm and 260/280 nm. Before the reverse transcription reaction, isolated RNAs were stored at −80 °C. For complementary DNA synthesis, 2 µg of total RNA was reverse-transcribed using random hexanucleotides and the High-Capacity cDNA Reverse Transcription kit (Thermo Fisher Scientific) following the manufacturer’s instructions. The reaction product was diluted to a final concentration of 20 ng/µL and stored at −20 °C until gene expression analysis.

### 4.6. Real-Time Quantitative Polymerase Chain Reaction (qRT-PCR) 

The qRT-PCR was carried out in a 7500 Fast Real-Time PCR System (Thermo Fisher Scientific) using a total volume of 10 µL according to the manufacturer’s instructions. We used TaqMan^®^ Fast Universal PCR Master Mix and assays based on hydrolysis probes (TaqMan^®^ Gene Expression Assays, Thermo Fisher Scientific), as they do not detect genomic DNA. Gene expression was performed for *CTLA-4* (Hs00175480_m1), *PDCD1* (PD1) (Hs00169472_m1), *CD274* (PD-L1) (Hs00204257_m1), *PDCD1LG2* (PD-L2) (Hs00228839_m1), *CD276* (B7-H3) (Hs00987207_m1), *JAK2* (Hs01078136_m1), and *FoXO1* (Hs01054576_m1) for cell lines, mammospheres, and co-cultures samples. As reference genes to normalize gene expression, we used both *β-ACTIN* (Hs99999903_m1) and *PUM1* (Hs00472881_m1). Depending on the assay type, a pool of RNA from the breast epithelial cell line 184A1 or the tumor cell lines was used as reference calibrator samples. No template controls were included in each reaction, and all experiments were duplicated. Relative changes in gene expression were calculated as fold change by the 2^−ΔΔCt^ method [111]. Results were analyzed with the 7500 software v2.0.6 (Thermo Fisher Scientific).

### 4.7. Statistical Analysis 

The Kolmogorov–Smirnov test was performed to define the distribution of the variables. For the differences in gene expression, parametric variables were compared using Student’s t-test, while non-parametric variables were compared using the Mann–Whitney U-test. In the co-cultures, in which repeated measurements were performed over time, the Student’s *t*-test for paired data and the Wilcoxon test were used to compare parametric and non-parametric variables, respectively. For our in vitro study, we define increased or decreased levels of any gene expression as those above or below a relative quantification value equal to 1. In all cases, a *p*-value of <0.05 was considered statistically significant. Statistical analysis was performed using the SPSS version-23 statistical software package.

## 5. Conclusions

In conclusion, we have found that *PDCD1* (PD1), *JAK2,* and *FoXO1* expression are highly variable in BC cell lines independently of adaptative immunity. In addition, intrinsic *CTLA-4*, *CD274* (PD-L1), and *PDCD1LG2* (PD-L2) expression could be essential for the pathogenesis of triple-negative BC cells, whereas *CD276* (B7-H3) expression would be necessary for luminal cell lines. Moreover, we have demonstrated the intrinsic expression of *CTLA-4, PDCD1* (PD1), *CD274* (PD-L1), *PDCD1LG2* (PD-L2), and *JAK2* in mammospheres derived from aggressive BC phenotypes. However, *CD276* (B7-H3) and *FoXO1* expression seem not to be directly involved in the acquisition and/or maintenance of tumor stem cell characteristics in vitro. Finally, we have determined that neoplastic cell-intrinsic *CTLA-4*, *PCDC1* (PD1), *CD274* (PD-L1), and *PDCD1LG2* (PD-L2) expression is dynamic and varies depending on BC immunophenotype and tumor-immune cell interactions. Therefore, these data confirm our hypothesis that the expression of some immunoregulatory genes depends on BC phenotype and stem cell properties. They further support the view that this expression is intrinsic to neoplastic cells and depends on the tumor microenvironment.

## Figures and Tables

**Figure 1 ijms-24-04478-f001:**
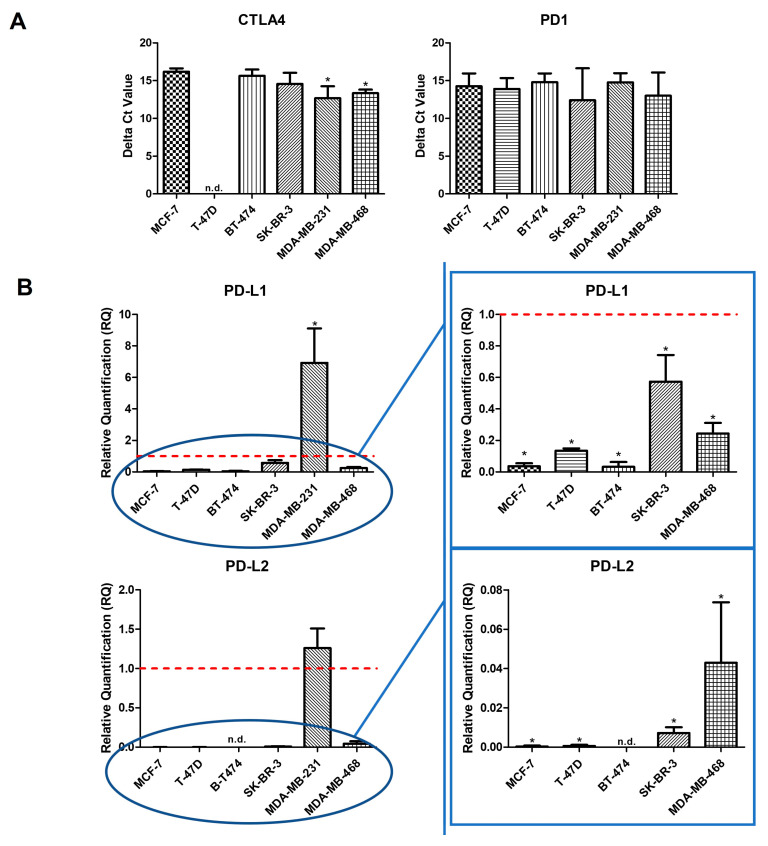
(**A**) Relative mRNA quantification of (**A**) *CTLA-4*, *PDCD1* (PD1), (**B**) *CD274* (PD-L1), *PDCD1LG2* (PD-L2), (**C**) *CD276* (B7-H3), *JAK2*, and *FoXO1* genes in BC cell lines. All experiments were performed in triplicate. Each graph shows the three replicates’ mean value and standard deviation. * *p* < 0.05; *** *p* < 0.001 compared to breast epithelium cell line 184A1 (red line). n.d., no data.

**Figure 2 ijms-24-04478-f002:**
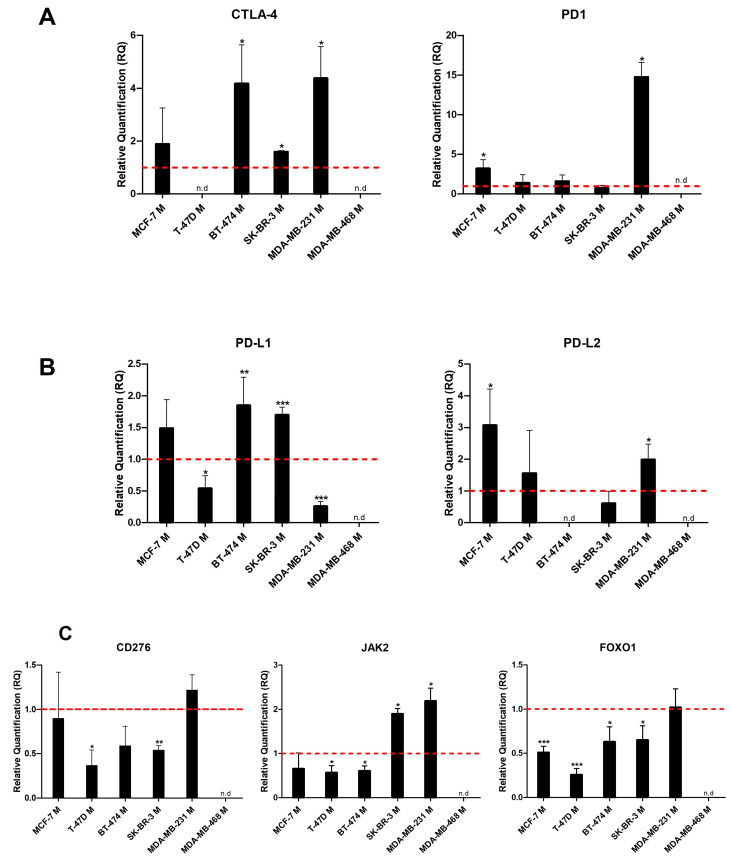
Relative mRNA quantification of (**A**) *CTLA-4*, *PDCD1* (PD1), (**B**) *CD274* (PD-L1), *PDCD1LG2* (PD-L2), (**C**) *CD276* (B7-H3), *JAK2*, and *FoXO1* genes in mammospheres (M) derived from breast cancer cell lines. All experiments were performed in triplicate. Each graph shows the three replicates’ mean value and standard deviation. ** p <* 0.05; *** p <* 0.01; **** p <* 0.001 compared to the corresponding BC cell line (red line). n.d., no data.

**Figure 3 ijms-24-04478-f003:**
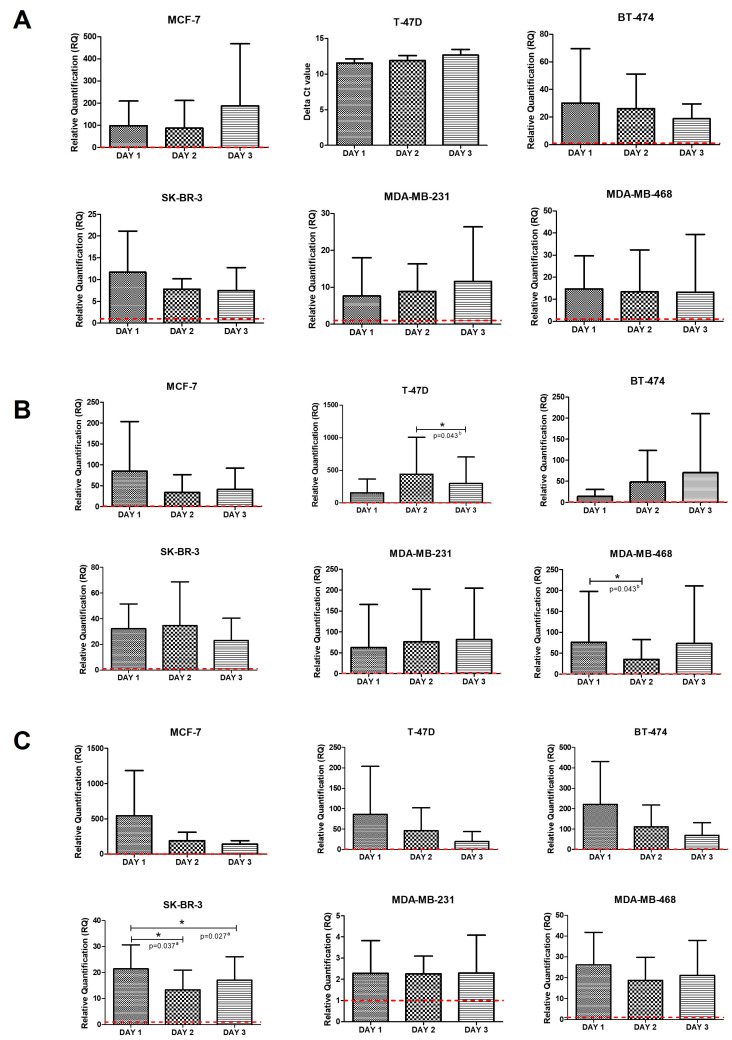
Relative mRNA quantification of (**A**) *CTLA-4*, (**B**) *PDCD1* (PD1), (**C**) *CD274* (PD-L1), (**D**) *PDCD1LG2* (PD-L2), (**E**) *CD276* (B7-H3), (**F**) *JAK2*, and (**G**) *FoXO1* genes in co-cultures of breast cancer cell lines and PBMCs. All experiments were performed in triplicate. Each graph shows the three replicates’ mean value and standard deviation. ** p <* 0.05; *** p <* 0.01 compared to the corresponding BC cell line (red line). ^a^ Student’s *t*-test for paired data; ^b^ Wilcoxon test.

**Figure 4 ijms-24-04478-f004:**
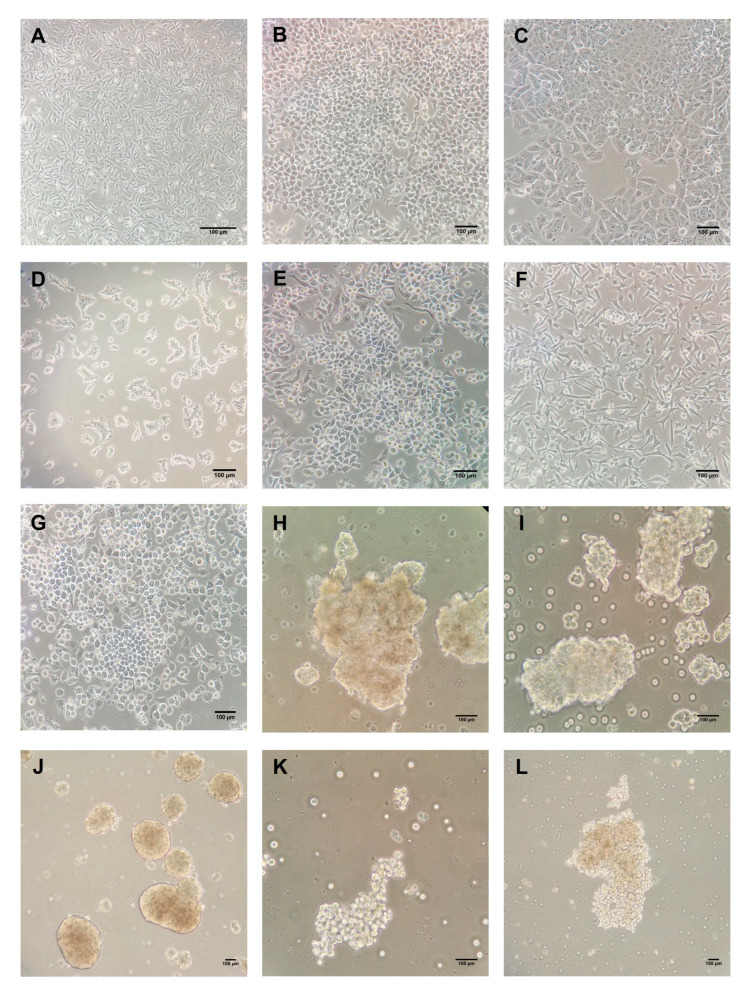
Breast epithelium ((**A**): 184A1; 20×) and breast cancer cell lines ((**B**): T-47D; 20×, (**C**): MCF-7; 20×, (**D**): BT-474; 20×, (**E**): SK-BR-3; 20× (**F**): MDA-MB-231; 20× and (**G**): MDA-MB-468; 20×). Mammospheres derived from breast cancer cell lines ((**H**): T-47D; 20×, (**I**): MCF-7; 20×, (**J**): BT-474; 20×, (**K**): SK-BR-3; 20× and (**L**): MDA-MB-231; 20×). Scale bar: 100 μm.

**Figure 5 ijms-24-04478-f005:**
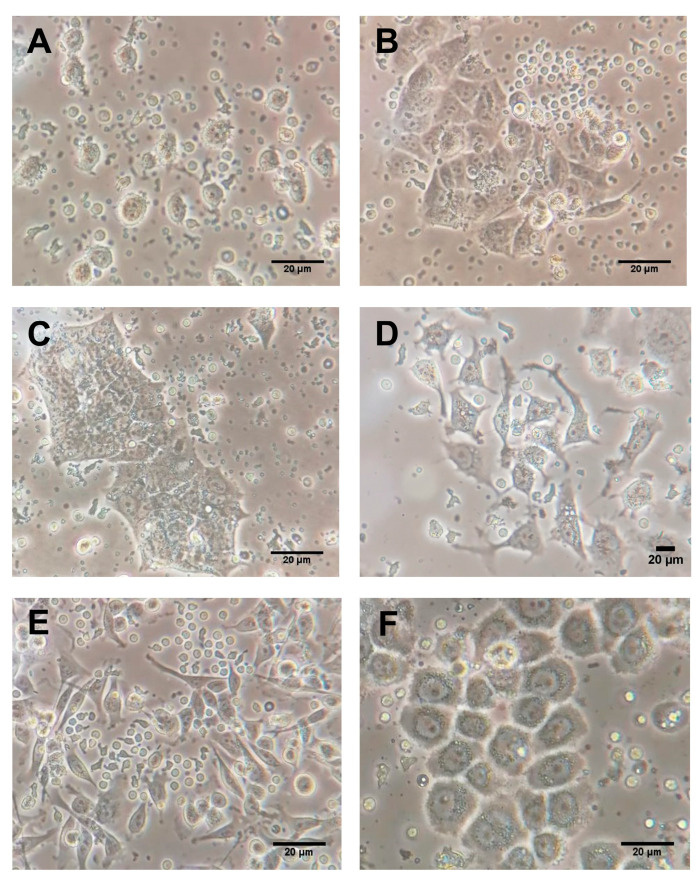
Co-cultures of breast cancer cell lines and peripheral blood mononuclear cells ((**A**): T-47D; 40×, (**B**): MCF-7; 40×, (**C**): BT-474; 40×, (**D**): SK-BR-3; 40× (**E**): MDA-MB-231; 40×, and (**F**): MDA-MB-468; 40×). Scale bar: 20 μm.

## Data Availability

The data presented in this study are available on request from the corresponding author.

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
