# Peer review of "Variable Intrinsic Expression of Immunoregulatory Biomarkers in Breast Cancer Cell Lines, Mammospheres, and Co-Cultures"

_ijms, 2023, doi:10.3390/ijms24054478_

Round 1

Reviewer 1 Report

The paper addresses a relevant and significant topic because a better understanding of immune biomarker expression in breast cancer subtypes is essential to responders vs non-responders.  Overall, the manuscript is well-written and the methodology is described in sufficient detail. The introduction provides relevant detail for readers to understand the significance of the study. The authors have made adequate effort to cite relevant and recent publications from this field.

Minor comments: 
In Figure 3, authors need to clarify in the figure legend what the superscript a and b, next to the p value, indicate.

Recommend authors to add a brief discussion of the success and failures of immunotherapy agents in breast cancer (example: FDA approval of PD-1 inhibitor pembrolizumab but failures of other checkpoint inhibitor agents) to contextualize their findings in relation to current clinical success of immunotherapy. That will help highlight the significance of this in vitro research.

Reviewer 2 Report

Review report IJMS

In the manuscript by Montoyo-Pujol et al, the authors compare gene expression profiles of several molecular markers in breast cancer cells. They analyzed various culture conditions to represent different subtypes of breast cancer as well and microenvironmental interactions. The authors predominantly use qRT-PCR based analysis for comparison of gene expression. Based on the manuscript submitted in its current form, there are several discrepancies that make the conclusions questionable.

1.       In Figure 1A, in the absence of a detectable expression of CTLA4 and PD1 in 184A1, it is suggested to normalize the data to one of the cell lines used. Based on the data presented there is no clear difference in gene expression the cell line panel analysed. The authors go on to claim that CTLA4 is overexpressed in TNBC cell lines which is not warranted.

2.       Figure 1C : Gene expression differences should be atleast 2 fold to be considered statistically different.

3.       The data provided are preliminary and based on one experimental analysis. This does not provide adequate information to make conclusions regarding breast cancer subtype variations.

4.       The data provided shows that gene expression is dynamic and is variable based on culture conditions. This makes the data hard to interpret and generalize. It also reduced the robustness for translational capacity.

Round 2

Reviewer 2 Report

Fold change values around 1.5 with a sizable error bar can arise from technical differences rather than biological differences. Hence the data need to be scrutinized more cautiously to make conclusions based on these small differences. The comparison of gene expression in various culture conditions is appreciated but the data does not reveal any clear subtype specific differences based on this comparison. 
